# Aspects Affecting Growth of Family Businesses

**Katarína Novotná \*, Zuzana Lušňáková**  **and Martina Hanová**

Faculty of Economics and Management, The Slovak University of Agriculture in Nitra, Tr. A. Hlinku 2,
94976 Nitra, Slovakia
\* Correspondence: xgurcikova@uniag.sk

**Abstract:** Family businesses form an inseparable part of each national economy. Therefore, it is necessary to pay significant attention to the aspects that will affect their growth and sustainability in the future. The main aim of this paper is to verify the implementation of key aspects within the entrepreneurial practice of family businesses that considerably affect their growth. Those aspects were identified while processing theoretical input data. On the basis of such knowledge, the research assumptions were determined, and their validity was subsequently verified through the Mann–Whitney U test, the non-parametric Kruskal–Wallis test and the $\chi^2$ test of independence. Afterwards, we were able to say that key determinants, such as a formal organisational structure, finance, a constituted family council and a proper code of ethics, have a considerable effect on the growth of family businesses in Slovakia. However, proper legislation that would be a fundamental pillar that family businesses may lean on in any of their operations and activities still needs to be enacted and put into practice.

**Keywords:** family business; management; manager; growth; development

## 1. Introduction

Family business has long been regarded as the oldest way of doing business throughout human history (Moresová and Sedliačiková 2018). Their start dates back to the ancient past when family businesses began carrying out their business activities in farming or agriculture (Schio Junior 2017). Family businesses are an important resource for economic development and are of great importance to the country's economy (Ferraro and Cristiano 2021). Family businesses play a key role in the national economy, participate in the country's GDP, considerably decrease unemployment and contribute to a country's wealth (Powell and Eddleston 2017). According to Slovak Business Agency (2020), family businesses are omnipresent in Slovakia, forming up to 99% of the total number of businesses. Therefore, their position vis-à-vis the state is indisputable. It is statistically proven that family businesses are mostly micro-, small- and medium-sized enterprises worldwide and that they play an important role in the country's economy and wealth (Wahyudi et al. 2021). These businesses mainly cover micro-, small and middle-sized enterprises. In Slovakia, the majority of family businesses were founded in the 1990s (Krošláková 2020), even though they were not clearly defined earlier than the year 2013. The current amendment to the Act no. 112/2018 Coll. on social economy and social enterprises and on amendments is supplementary to certain acts ("Social Economy Act") that should properly amend and govern family entrepreneurship and family businesses (Ildža 2021). Nowadays, ensuring the development of either new or already existing family businesses and the identification of the aspects of their development is urgently needed. Our main aim is to identify and verify the importance and significance of key determinants affecting the growth of family businesses in Slovakia. The meaning of a family business is not precisely and expressly defined in many countries, not even in scientific or academic literature, despite being regarded as the basis for the modern world economy (Schio Junior 2017). One of the main reasons for the insufficient establishment of family businesses in the economic sciences is

due to the lack of a definition of the term of family business in many countries (Harms 2014). World literature has brought various definitions of family business. Churchill and Hatten (1997) viewed a family business as a business through succession, Shanker and Astrachan (1996) described a family business as a company meeting strict criteria to be categorised as a family business. These criteria mostly relate to the capital and ownership structure, the size of a business, the number of family relatives being employed in a business, the number of generations, etc. Surdej and Wach (2012) defined a family business as a company owned by a family without limitations to its ownership structure.

According to the Slovak Business Agency (2020), the key characteristics of family businesses are as follows:

1.  A natural person that established a family business keeps most decision-making rights, or a natural person who acquired the company-registered capital, or one which is owned by other direct family relatives, such as a spouse, parents, children or an heirs' direct descendants.
2.  Most decision-making rights are either direct or indirect.
3.  At least one representative of the family or relatives formally takes part in company management.
4.  The person who either established or acquired the business (its share capital) or their families or descendants have 25% of decision-making rights as determined by their share capital.

When it comes to their legal form, current family businesses tend to be mostly freelancers, followed by limited companies and, to a smaller extent, joint-stock companies (Rydvalová et al. 2015). A family business and ownership may have various definitions. The main goal of a family business is to reach cohesion and equality between company management and ownership, while accomplishing key company goals (Poza 2010). One of the main issues family managers have to deal with is, besides succession and generational change, ensuring the growth of their business (Mura 2020). The strategic behaviour of family businesses is more conservative in contrast to non-family businesses. Family businesses are perceived as more stable entities, which gives them lots of advantages and makes them more sustainable (Németh 2020), with their growth being more modest than that of non-family businesses. However, paradoxically, Lee (2006) assumes that family businesses tend to grow much faster than non-family ones. Owners of family businesses are trying to attain a long-term economic performance of their business (Zellweger et al. 2013). Financial performance is the result of carrying out the business activities of a family business. It is a measure of the impact that companies have on business development in their day-to-day activities, and it reflects the company's operating results and standards for assessing the financial position in a particular accounting period through its financial statements. Financial performance is a measure of the results of corporate policies and operations in monetary terms (Li 2022). Insolvency is one of the key determinants threatening company financial stability (Gurčík 2011). The promoters of family businesses should ensure there are enough financial resources (Meier and Franková 2019). A scenario in which family managers do not have sufficient financial resources to establish or manage their company and are forced to look for foreign capital, such as bank loans, can easily happen. According to the research by Moresová et al. (2021), most family businesses do not divide the financial resources of the company and that of the family, which was proven in the research by Zellweger et al. (2013). The goals of family businesses do not only focus, as stated by Belenzon et al. (2015), on profit maximising, but on other business goals, such as family values, providing employment to their family relatives or keeping the family united. Management goals might often have higher a significance to family businesses than financial ones (Chrisman et al. 2010). Paying attention to non-financial goals and possessing human resources will lead to the success of a business that wants to stay on the market (Šajbidorová et al. 2016). According to Puiu et al. (2022), financial capital is precisely the point from which a business develops, while Toubes and Araújo-Vila (2022) consider that financial capital is not enough, but that it is also necessary to have high-quality human resources or social capital

to support family businesses. Having specific management apparatuses brings efficient business results. Adopting a code of ethics, having a properly constituted family council or an informal organisational structure of a family business are, according to Rivo-López et al. (2021), regarded as determinants paving the way for business growth. Mura et al. (2021), while observing V4 countries, stated: "The introduction of Code of Ethics is strongly recommended, as unethical behaviour is frequent, while Code of Ethics is easy to implement". Zsigmond et al. (2021) investigated the effect of the company size on the existence of ethical institutions (as a code of ethics) in the case of Slovakian enterprises. They stated that "56.5% of the respondents answered that there are not ethical institutions in their company, while 43.5% reported that at least one ethical institution is operating in the company". A family council is a formally constituted institution consisting of family members created for the purpose of accomplishing various tasks and sorting out various issues. One of the main goals is to plan a successorship, innovations and changes in the company management, as well as ensuring that company and family goals match (Machek 2017). A family council serves as an advisory body for the whole company (Martel 2017). A family council is typical for large enterprises (Fabianová and Janeková 2020), although the expert Matwij (2021) states that a family council is necessary for all types of family businesses regardless of their size. A code of ethics is a document containing guidelines and best practices for employees' ethical behaviour (Smith et al. 2020). According to Payne et al. (2019), a code of ethics belongs among the formal tools of company management guaranteeing employees' ethical behaviour. A code of ethics must be properly implemented into the company and regularly updated to reflect company changes. It is an important management tool for each family business, while being mostly used in middle-sized and large enterprises (Hogenbirk and van Dun 2021). Another important aspect of family businesses, according to Gallo (2017), is an effectively prepared and implemented business organisational structure. The organisational structure of family businesses might sometimes be more complex than in non-family businesses (Mura and Kajzar 2019). The formal organisational structure is proposed for the purpose of efficient company management with employees accomplishing their goals at particular levels. However, this management tool does require innovations. Changing the organisational structure must be carried out perfectly because by neglecting mistakes, the company may easily end up in a crisis (Čepelová 2005). Based upon the analysis of theoretical knowledge from domestic as well as foreign authors (Martel 2017; Fabianová and Janeková 2020; Hogenbirk and van Dun 2021; Gallo 2017; Payne et al. 2019; Zellweger et al. 2013, etc.), the three research assumptions were formulated and subsequently verified through the analysis:

RA 1: Family businesses with a formal organisational structure demonstrate higher management efficiency.

RA 2: The size of the family business determines whether the family business has a family council and a code of ethics.

RA 3: If the management of family businesses differentiate ownership and management, they also divide company and family finance.

## 2. Methodology

The main aim of the paper is to verify implementation of key aspects significantly affecting the growth of family businesses in Slovakia, which were identified while processing the theoretical data. The empirical research was carried out to obtain input data. The questionnaire research was conducted in family businesses across Slovakia by means of electronic communication. Due to the fact that Slovak legislation lacks any enactment on family businesses, as stated above, it was not possible to obtain a database of family businesses or determine the representative size of the research sample. We obtained the said data from the Finstat database (the online platform containing data about Slovak businesses—the database of businesses and their course of events, company profiles, financial reports and datasets) of which we managed to generate micro-, small- and middle-sized enterprises. The number of employees was the main criterion for determining company

size. By matching the surnames in the company or supervisory board, we could identify whether the business in question was or was not family run. Another criterion for selecting the businesses was the most common definition of traits of a family business based upon the non-binding specification created by the expert group of the European Commission from 2009:

"This definition says that the family business can be defined as a business of any size if:

- most decision-making rights are vested in the persons who established the business or the persons who acquired capital share in the business or in the ownership of spouses, parents, children or direct descendants of children;
- most decision-making rights are either direct or indirect;
- at least one family representative or his/her relative is formally involved in the company management;
- in case of businesses listed on the stock exchange, the person who established the business or acquired its share (or the person's/persons' family or relatives) owns min. 25% of decision-making rights according to their capital share".

The questionnaire was sent from January to April 2022 to 412 family businesses from all over Slovakia. There was an 18% return with 72 questionnaires having been properly filled in. The questionnaire was made of two parts. The first part contained identification data where we identified the region where the company was operating, the field of business, the year of establishment, the generational chronology and the company financial situation for the past two years.

Based upon the identification traits, we tried to specify the representation of family businesses across the regions. The sample consisted of family businesses from all Slovak regions. When it came to their size structure, micro-businesses accounted for 50%, small businesses for 35%, middle-sized businesses for 12% and large ones for only 3%. According to the SBA survey, (Slovak Business Agency 2021), micro-, small- and medium-sized enterprises accounted for 97.3% of the total number of active business entities in Slovakia. Based on this survey, we can confirm that our sample of family businesses matched the sample of the SBA business research.

Most family businesses in Slovakia which took part in the research were established after 2011. The fewest number of family businesses operated in trading, whereas the majority from the selected sample conducted their business in production. Up to 71% of businesses were still managed by the first generation.

The last identification trait was the company financial situation for the past two years. The results demonstrated that the financial situation of 42% of family businesses has not considerably changed, while 20% of the respondents saw their financial situation deteriorate.

The second part of the questionnaire had 9 statements where the respondents could express their level of agreement/disagreement on the five-point Likert Scale: (1) strongly disagree; (2) disagree; (3) neither agree nor disagree; (4) agree; and (5) strongly agree. The input data from the empirical research allowed us to create the numeric matrix in MS Excel to be further processed and evaluated by means of mathematical–statistical methods. According to the type of survey questions, we used association and multi-sample analysis to verify the established hypotheses. The statistical methods that were applied to assess and verify scientific assumptions included the Mann–Whitney test, the Kruskal–Wallis test and the Chi-square test of independence. Applied statistical analyses were performed in Statgraphics Centurion 18. The results were interpreted, and the recommendations were formulated.

## 3. Results and Discussion

On the basis of the theoretical analyses of the findings, three research assumptions were formulated within our research. Based upon these assumptions, the statistical hypotheses

were determined, and they were subsequently verified by means of mathematical and statistical methods.

Further to the statements of various authors on the formal organisational structure as one of the key management tools within a company with a considerable impact on the company efficiency, the following research assumption was determined:

RA1: Family businesses having a formal organisational structure demonstrate higher management efficiency.

This assumption was verified by the statistical method of the $\chi^2$ test of independence, with the Mann–Whitney U test for the confirmation of the results and clear graphic displays showing the respondents' answers on a five-point Likert Scale.

Figure 1 below graphically shows the relation between using a formal organisational structure and the level of management efficiency of family businesses. The respondents' answers on the five-point Likert Scale show that managers of 51 family businesses had a proper formal organisational structure, while 21 of them had none. Family businesses with a proper organisational structure strongly confirmed that this management tool has a positive impact on the company efficiency (57% agreed and 25% strongly agreed). Family businesses without a proper organisational structure did not agree with its substantial impact on company efficiency (62% family businesses made a neutral statement and 39% disagreed).

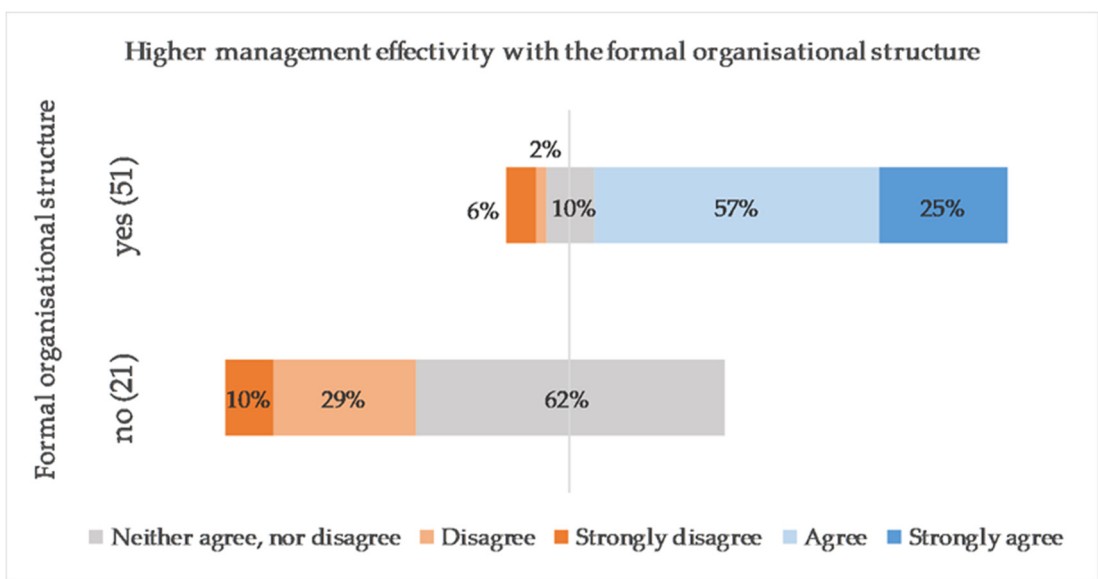

**Figure 1.** Relation between using a formal organisational structure and level of management efficiency.

The validity of the scientific assumption was verified by applying the statistical method of $\chi^2$ test of independence. We found out that the *p*-value (0.00) was lower than the level of significance of 0.05; therefore, we could confirm the above research assumption. Subsequently, the result was verified by the Mann–Whitney U test, where the *p*-value was $3.16075 \times 10^{-8}$, (lower than 0.05 and 0.001 and W = 108.5), which confirms our previous statement and the validity of the scientific assumption.

Another research assumption was formulated on the basis of theoretical knowledge from Fabianová and Janeková (2020), Martel (2017), Zsigmond et al. (2021) and Payne et al. (2019). They all assumed that a family council is a formally constituted institution which ensures the management efficiency of a family business. It is a management tool typical for large businesses, the same as a code of ethics. For many micro-businesses with a lower number of employees, a code of ethics seems to be rather unnecessary.

In light of the above, the following research assumption was determined:

RA2: The size of the family business affects whether the family business has a family council or a proper code of ethics.

To verify the validity of the above research assumption, the $\chi^2$ test of independence was applied, along with the extended Mann–Whitney U test and the non-parametric Kruskal–Wallis test.

The relation between the company size and the existence of a family council and code of ethics is shown in Table 1 below. The fully functional family council was present in 65% of family businesses taking part in the research but lacking in 35% of them. A code of ethics was present in 40% of all asked businesses but in 60% of the total number of respondents, a code of ethics was missing (mostly in micro and small family businesses).

**Table 1.** Cross tabulation—company size on the existence of a family council and code of ethics.

| Company Size | | Family Council | | Code of Ethics | | Total |
| | | Yes | No | Yes | No | |
| --- | --- | --- | --- | --- | --- | --- |
| Micro | Number | 21 | 15 | 13 | 23 | 36 |
| | Frequency | 29% | 21% | 18% | 32% | 50% |
| Small | Number | 16 | 9 | 9 | 16 | 25 |
| | Frequency | 22% | 13% | 13% | 22% | 35% |
| Middle-sized | Number | 8 | 1 | 5 | 4 | 9 |
| | Frequency | 11% | 1% | 7% | 6% | 13% |
| Large | Number | 2 | 0 | 2 | 0 | 2 |
| | Frequency | 3% | 0% | 3% | 0% | 3% |
| Total | Number | 47 | 25 | 29 | 43 | 72 |
| | Frequency | 65% | 35% | 40% | 60% | 100% |

Based upon the demonstration of respondents' answers, the company size has no effect on the existence of a family council or code of ethics.

In order to determine the effect of the company size on the existence of a family council, the $\chi^2$ test of independence was applied. The results (Table 2) show that there was no statistically significant difference in the respondents' answers. This independence was confirmed by the subsequent result of the Kruskal–Wallis test, with a *p*-value of 0.261 and testing statistics value of 4.005.

**Table 2.** Association analysis—company size by family council and code of ethics.

| Test | | Company Size by Family Council | Company Size by Code of Ethics |
| --- | --- | --- | --- |
| Chi-square test | Statistic | 4.061 | 4.289 |
| | *p*-Value | 0.255 | 0.232 |
| Kruskal–Wallis test | Statistic | 4.005 | 4.299 |
| | *p*-Value | 0.261 | 0.238 |
| Measures of Association | | | |
| Pearson's R | Statistic | 0.220 | 0.187 |
| | *p*-Value | 0.063 | 0.115 |
| Kendall's Tau | Statistic | 0.187 | 0.136 |
| | *p*-Value | 0.097 | 0.227 |
| Spearman Rho | Statistic | 0.199 | 0.143 |
| | *p*-Value | 0.099 | 0.234 |

For the relation between the size of a family business and the existence of a code of ethics, the application of the $\chi^2$ of test of independence led to a *p*-value (0.2319) higher than the significance level of 0.05. Therefore, we cannot definitely determine significant differences in respondents' answers. The same applies for the relation between the company

size and the existence of a family council, which we determined by the confirmation of the Kruskal–Wallis test with a *p*-value of 0.238 and a testing statistics value of 4.229, higher than 0.005.

The findings for both determinates were also confirmed by the measures of association, also listed in Table 2. They indicate a low-to-moderate strength of association and are statistically insignificant.

The third research assumption was determined on the basis of the research results by Moresová et al. (2021), as well as Zellweger et al. (2013), who assumed that most family businesses do not divide financial resources of the company and that of the family.

RA3: If the management of family businesses differentiate ownership and management, they also tend to divide company and family finance.

The above assumption and determined statistical hypotheses were verified by the Mann–Whitney U test.

Figure 2 below shows the respondents' answers on the five-point Likert scale. It is obvious that respondents' answers varied considerably. The managers of family businesses dividing the company ownership and management either strongly agreed (42%) or agreed (32%) on making a clear line between company and family finances (47% strongly agreed and 28% agreed).

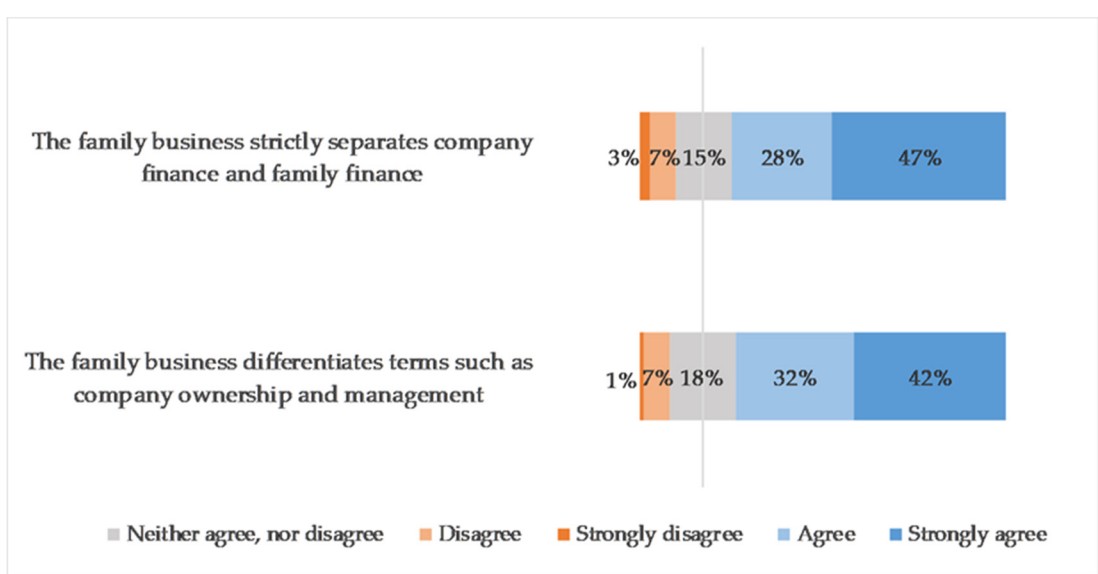

**Figure 2.** Structure of respondent's opinions on separation of company and family finance and differentiating company ownership from management.

The statistical independence between the answers to the questions on differentiating between management and ownership, as well as separating company and family finances, was verified by the Mann–Whitney U test, with a *p*-value (0.635, W = 2480.0) higher than the level of significance (0.05). Therefore, we demonstrated no statistically significant difference in the respondents' answers to the questions on differentiating ownership and management and separating company and family finances.

The third research assumption has been confirmed: if the management of family businesses divide ownership and management, they also tend to separate family and business finance.

## 4. Discussion

Based upon the above results, we can easily say that family businesses with a formal organisational structure definitely demonstrate higher management efficiency. The research assumptions confirmed by the research also matches the opinion presented by Gallo (2017),

who stated that the formal organisational structure, in the respondents' opinions, affects the management efficiency of family businesses. Research into family businesses in Slovakia also confirmed the significance and the role organisational structure plays in 71% of businesses. On the basis of research works of various authors, we also have to see the formal organisational structure as a key aspect of company management, ensuring company growth and sustainability. According to Čepelová (2005), the organisational structure is expected to be formal, functional, clear and properly implemented. We recommend that the formal organisational structure be implemented in those family businesses which still lack this form of company management, while advising this formal organisational structure as a key aspect of company management, ensuring its growth. The organisational structure must be fully functional, clear and properly implemented.

Despite the statements represented by Fabianová and Janeková (2020), Martel (2017) and Payne et al. (2019), our next research results showed that company size has no effect on the existence of a family council and a code of ethics. According to the research results of Zsigmond et al. (2021), there is also a relationship between the existence of ethical institutions and the size of the company. Another expert Matwij (2021), however, states that constituting a family council and a proper code of ethics is deemed crucial for successful company management and development. A family council and a code of ethics have always been essential management aspects for all sizes of family businesses, which has also been confirmed by our research which shows that 65% of the family businesses in question have a proper family council and 40% of them do have a code of ethics, which is regarded as a key aspect for ensuring their functioning and development.

Our findings in the area of "dividing ownership and management and separation family and business finance at the same time" match with those of Poza (2010), who affirmed that, in properly managed family businesses, the manager tries to establish an optimal harmony and balance, both in the workplace and the family. It is recommended to differentiate between management and ownership in family businesses as confirmed by our research, where 42% of family businesses divide the role of the owner and that of the manager. In the first phase of the company life cycle, the owner often matches with the manager, but in the following stage, this function is usually clearly separated (Machek 2017). Both positions have specific goals to be accomplished while keeping mutual respect. It is advisable to draw up the formal organisational structure of the family business, create a family council or form a code of ethics. Finance is another important aspect that has been identified as a key aspect to the growth of family businesses. The results of the analysis confirmed that the respondents tended to separate company and family finance. According to the results of the research conducted by Moresová et al. (2018), it is obvious that, within the internal determinants of family businesses, the area of finance is problematic—especially the lack of equity capital, where up to 50% of respondents identified it as the most negative determinant. Toubes and Araújo-Vila (2022) claim, however, that in the case of insufficient capital, company funds usually come from the family and not from financial institutions or banks. They assume that family and company finances are not differentiated, even though our research into Slovak family businesses clearly shows that 47% of businesses tend to divide finance.

## 5. Conclusions

In Slovakia, family businesses mainly refer to micro-, small and middle-sized enterprises. According to the Slovak Business Agency (2021), family businesses contribute by up to almost 99% of the total number of business entities registered in Slovakia. It is therefore essential to pay more attention to family businesses from the perspective of the state especially, through legal enactments governing family entrepreneurship in Slovakia, which will be beneficial not only to the managers and owners of family businesses themselves, but also to the whole scientific and research community.

The main aim of this paper was to verify the implementation of key aspects in the practice of family businesses which have been identified through the processing of the theoretical data; for example, the aspects that have a considerable impact on their growth.

By analysing this theoretical knowledge, we found out that the aspects affecting company growth mainly include the formal organisational structure, as it directly relates to increasing the management efficiency of the specific company. A family council and a code of ethics are among other key determinants influencing business growth. The company size has not been identified in our research as a factor for the existence of a family council or a code of ethics. However, the research of other authors shows that any kind of family business should have its own family council and hold regular meetings to make decisions about needs, proposals, changes or innovations or any other important business issues. Many family businesses do not differentiate ownership and management, but ensuring the cohesion and common accomplishment of goals is vital for company growth, both on the part of family members and family-business management. Finances and their origin are a key aspect for the growth of each company. The results of the research expressly confirmed that family businesses divide family and business finance for fear of failure when it comes to investing family finance.

The legal enactment of family business matters is nowadays of the utmost importance to provide a fundamental pillar that family entrepreneurs may rely on, as well as to ensure easier access to scientific researchers for the purpose of conducting research in the given field. This definition must be clear, concise, expressive and functional. In Slovakia, we still lack a scheme of support for family businesses ("de minimis" state aid) the main goal of which would be to promote family businesses.

The key determinants affecting the business activities of family businesses include insufficient state support, legal loopholes, extreme bureaucracy, high taxes levies, and many more. The measures to be taken to promote the growth of family businesses may include the following: more efficient state support, state financial aid, promoting training and lectures for employees, taking legislative steps, etc. Economic, technological and legislative factors are the most important macro-environmental factors family businesses have observed.

Limitations and future directions: As part of obtaining information and entering data into the analyses, we faced many limitations. One of them was the time and financial complexity of the data collection through a questionnaire survey during the ongoing consequences of the pandemic situation, which affected the management of family businesses. They had to, and especially wanted to, focus on ensuring survival. Additionally, it was often a challenge to convince family businesses, especially smaller ones, of the importance of perceiving and solving the determinants identified by our research. The further direction of research activities will be aimed at expanding the areas of aspects that are identified by scientists as those that most strongly influence the sustainability of family businesses in Slovakia. At the same time, we are convinced that a qualitative and humane approach to research and a sensitive perception of the needs of family-business owners, who have an irreplaceable place within business entities, will bring quality outputs and prerequisites for the formulation of solutions.

**Author Contributions:** Conceptualization, Z.L. and K.N.; methodology, Z.L., M.H. and K.N.; software, M.H.; validation, K.N. and M.H.; formal analysis, Z.L. and K.N.; investigation, Z.L. and K.N.; resources, K.N.; data curation, M.H.; writing—original draft preparation, K.N. and Z.L.; writing—review and editing, K.N., Z.L., M.H.; supervision, Z.L. All authors have read and agreed to the published version of the manuscript.

**Funding:** This research was funded by the Scientific Grant Agency of the Ministry of Education, Science, Research and Sports of the Slovak Republic and the Slovak Academy of Science, Slovakia, grant number VEGA 1/0490/21: "Factors of success in the process of succession in a small, medium and micro family business in Slovakia. Qualitative and quantitative approaches to analysis and solutions".

**Informed Consent Statement:** This was waived for this study due to the fact that we used anonymous data that were not traceable to individuals at any time.

**Data Availability Statement:** Not applicable.

**Conflicts of Interest:** The authors declare no conflict of interest.

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
