# Peer review of "Aspects Affecting Growth of Family Businesses"

_economies, doi:10.3390/economies10100256_

Round 1

Reviewer 1 Report

Work does not make a significant contribution to the science and practice of family businesses in Europe and in the world.

The research questions posed do not solve new problems and were presented around the world in various thematic configurations.

Nevertheless, the article may be interesting for readers of Slovakia, the Czech Republic and the nearest neighboring countries due to its regional character.

The presentation of the results is qualitative and you should consider changing the presentation of the results and abandoning the drawings that are of little use.

The literature on the subject is incomplete, but it can be considered sufficient. There are errors in the alphabetical arrangement of the entries in line 407, 458, 460

Author Response

Dear,
thanks for the comments, we have corrected the manuscript and did the best we could. I am sending a document with a comment in the attachment.

Sincerely,
Ing. Katarína Novotná

Reviewer 2 Report

Dear Authors,

find my comments attached!

All the best!

Author Response

(The authors gave the same response as above.)

Reviewer 3 Report

The paper is well constituted and well developed. As a matter for improvement, it would be advisable to incorporate certain explanatory aspects of the methodological development that has served as a basis for the development of the work.

Author Response

(The authors gave the same response as above.)

Round 2

Reviewer 2 Report

Dear Authors,

thank you for correcting the errors based on my comments. The article has been greatly improved. 

The last thing I discovered is that I think the "Limitations and future directions" should not be with bold. Maybe with italics (as a sub-chapter). Also, the last sentence of the paragraph should not be with bold.

But maybe the editing team will fix it - in case of accepting!

Thank you again and all the best!